# Hierarchical modelling of factors associated with the practice and perpetuation of female genital mutilation in the next generation of women in Africa

**Adeniyi Francis Fagbamigbe**[1,2]*, **Imran Oludare Morhason-Bello**[3,4,5], **Yusuf Olushola Kareem**[4], **Erhabor Sunday Idemudia**[6]

1 Faculty of Public Health, Department of Epidemiology and Medical Sciences, College of Medicine, University of Ibadan, Ibadan, Nigeria, 2 Division of Population and Behavioural Sciences, School of Medicine, St Andrews University, St Andrews, Fife, United Kingdom, 3 Faculty of Clinical Sciences, Department of Obstetrics and Gynaecology, College of Medicine, University of Ibadan, Ibadan, Nigeria, 4 Institute for Advanced Medical Research and Training, College of Medicine, University of Ibadan, Ibadan, Nigeria, 5 Centre for Population and Reproductive Health, College of Medicine, University of Ibadan, Ibadan, Nigeria, 6 Faculty of Humanities, North-West University (MC), Mafikeng, South Africa

* franstel74@yahoo.com

**Data Availability Statement:** The data supporting this article is available at: http://dhsprogram.com/

## Abstract

Despite a total prohibition on the practice of female genital mutilation (FGM), young girls continue to be victims in some African countries. There is a paucity of data on the effect of FGM practice in two generations in Africa. This study assessed the current practice of daughters' FGM among women living in 14 FGM-prone countries in Africa as a proxy to assess the future burden of FGM in the continent. We used Demographic and Health Surveys data collected between 2010 and 2018 from 14 African countries. We analyzed information on 93,063 women-daughter pair (Level 1) from 8,396 communities (Level 2) from the 14 countries (Level 3). We fitted hierarchical multivariable binomial logistic regression models using the MLWin 3.03 module in Stata version 16 at p<0.05. The overall prevalence of FGM among mothers and their daughters was 60.0% and 21.7%, respectively, corresponding to 63.8% reduction in the mother-daughter ratio of FGM. The prevalence of FGM among daughters in Togo and Tanzania were less than one per cent, 48.6% in Guinea, with the highest prevalence of 78.3% found in Mali. The percentage reduction in mother-daughter FGM ratio was highest in Tanzania (96.7%) and Togo (94.2%), compared with 10.0% in Niger, 15.0% in Nigeria and 15.9% in Mali. Prevalence of daughters' FGM among women with and without FGM was 34.0% and 3.1% respectively. The risk of mothers having FGM for their daughters was significantly associated with maternal age, educational status, religion, household wealth quintiles, place of residence, community unemployment and community poverty. The country and community where the women lived explained about 57% and 42% of the total variation in FGM procurement for daughters. Procurement of FGM for the daughters of the present generation of mothers in Africa is common, mainly, among those from low social, poorer, rural and less educated women. We advocate for more

data/available-datasets.cfm, which can be downloaded upon request from the DHS website.

**Funding:** The author(s) received no specific funding for this work.

**Competing interests:** The authors have declared that no competing interests exist.

**Abbreviations:** CI, Confidence Intervals; DHS, Demographic and Health Surveys; FGM, Female Genital Mutilation; HDI, Human Development Index; IGLS, Iterative Generalized Least Squares; IRB, Institutional Review Board; ICC, Intra-Class Correlation; MQLL, Marginal Quasi-Likelihood Linearization; MOR, Median Odds Ratio; ORs, Odds Ratios; SES, Socio-Economic; VIF, Variance Inflation Factor; VPC, Variance Partition Coefficient.

context-specific studies to fully assess the role of each of the identified risk factors and design sustainable intervention towards the elimination of FGM in Africa.

## Introduction

Female genital mutilation (FGM) is an old socio-cultural practice that is usually carried out among girls and women, mainly, in Africa and Asia, and among immigrants in western countries [1–4]. FGM involves partial or complete removal of the external genitalia of women for non-therapeutic reasons. This cultural practice is associated with sexual dysfunction, obstetric complications and mental health problems among survivors [5–7]. Over the years, international agencies–WHO, UNICEF and development partners have supported governments to formulate policies, enact laws, institute programmes and interventions to prevent the practice of FGM [8–10]. For example, the WHO has encouraged countries to enact laws that prohibit all forms of FGM, and that this practice should constitute a form of gender-based violence as well as a violation of human right [8,11].

Despite restriction to the practice of FGM, the report of the number of girls that are suffering from FGM continue to soar in Africa [4,12]. It is estimated that more than 200 million women had suffered from FGM, and the majority are young girls living with the scars of FGM [13]. Apart from these, many more of these girls and women also have additional medical problems ranging from childbirth and urological complications, psychosomatic problems and permanent disabilities with poor quality of life [13–15]. In 2018, the WHO estimated that 1.4 Billion US$ will be required per annum to manage health complications associated with FGM, and if nothing is done, this might increase to 2.3 Billion US$ per annum in 30 years (2047) [13].

The survival and increasing practice of FGM in Africa are associated with different drivers: the strong respect for culture, low status of women in the society and religious belief are key drivers that promote FGM in different settings [16–18]. Women, mainly, in Africa are the victims of cultural belief; FGM is believed to be an essential cultural rite that a woman must accept to attain and maintain a respectable status in the society [7,19]. Mothers are expected, as part of their role in the family and society, to support and promote cultural practices, including having FGM for their daughters [19]. Studies from many African countries have shown that women irrespective of their education, wealth and location, would not mind that FGM should be performed on them and their daughters [20–23]. The favourable disposition towards FGM is further enhanced among women, especially, in some Muslim settings, where such practice is viewed as a religious rite [17,24]. Studies in Egypt and Burkina Faso showed that 60% and 32% of women, respectively, would support FGM to be performed on their daughters [22,25].

In 2014, a study in Africa had previously reported on the practice of FGM within families in Burkina Faso, Senegal and Egypt [26]. The study demonstrated that FGM was more likely to be reported among daughters of mothers who had FGM relative to those whose mothers were not [26]. The study adjusted for some key confounders such as religion, place of residence and number of daughters in the family [26]. However, the three countries have a relatively high population of Muslims compared to other countries with possible high vulnerability to FGM practice[26]. Also, both Burkina Faso and Senegal are French-speaking countries with the plausibility of having similar cultural norms [26]. To appreciate the extent of the burden of FGM in the future generations of women in Africa, it is important to understand the risk of

this social norm in different settings that cut across language, religion, culture and nationalities in the continent among women of reproductive age. Besides, there is a need to update the body of knowledge and stakeholders on the burden of FGM among children who to make necessary intervention plans, as these children are the women of tomorrow. This study assessed the practice of daughters' FGM among women living in 12 countries in Africa as a proxy to estimate the future burden of FGM in the continent. Also, the analyses compared the prevalence and the risk of FGM among daughters of women with or without FGM interviewed during the survey.

## Materials and methods

### Data source

The most recent data collected during the Demographic and Health Surveys (DHS) conducted between 2010 and 2018 in African countries was used. The DHS data are cross-sectional and nationally representative conducted among women of reproductive age (15 to 49 years) in the respective countries. We identified only 14 countries with data set on FGM for both the respondents and their daughters. Countries without module on FGM for the respondents and their daughters were dropped. The countries included in the analysis are Burkina-Faso, Chad, Cote d'Ivoire, Egypt, Ethiopia, Guinea, Kenya, Mali, Niger, Nigeria, Senegal, Sierra Leone, Tanzania, and Togo. The details of the year of survey and the number of daughters whose FGM data were reported, the prevalence of FGM among both the mothers and daughters are shown in Table 1.

### Sampling

Typically, the DHS adopted multistage cluster probability sampling methods to select eligible respondents in each of the countries. Within each state/district/region in each country, the enumeration areas, also known as clusters, were selected. The clusters are the primary

**Table 1. Data characteristics and prevalence of FGM among respondents and their daughters.** DHS 2010–2018.

| Country | Year of survey | Number of Neighbourhoods | Number of women | Prevalence of Mothers' FGM | Prevalence of daughters' FGM | Mother-Daughter % reduction |
|---|---|---|---|---|---|---|
| Nigeria | 2018 | 1304 | 8291 | 36.0[33.5–38.6] | 30.6[28.0–33.3] | 15.0 |
| Kenya | 2014 | 1580 | 7226 | 26.4[24.6–28.2] | 3.6[3.1–4.3] | 86.4 |
| Tanzania | 2015 | 608 | 5912 | 15.2[12.8–18.1] | 0.5[0.4–0.8] | 96.7 |
| Senegal | 2017 | 400 | 7684 | 28.7[25.3–32.3] | 15.0[12.6–17.7] | 47.7 |
| Ethiopia | 2016 | 624 | 3829 | 79.9[76.2–83.1] | 23.4[20.3–26.8] | 70.7 |
| Burkina Faso | 2010 | 573 | 10064 | 82.1[80.1–83.9] | 17.5[16.0–19.1] | 78.7 |
| Guinea | 2018 | 401 | 5559 | 97.0[96.1–97.7] | 48.6[46.5–50.7] | 49.9 |
| Cote d'Ivoire | 2012 | 351 | 4864 | 44.8[40.8–48.8] | 13.1[11.3–15.2] | 70.8 |
| Mali | 2018 | 341 | 3147 | 93.1[91.4–94.5] | 78.3[76.0–80.5] | 15.9 |
| Niger | 2012 | 137 | 3244 | 5.0[3.4–7.2] | 4.5[3.2–6.2] | 10.0 |
| Sierra Leone | 2013 | 266 | 9759 | 96.6[95.7–97.3] | 31.4[29.9–32.9] | 67.5 |
| Togo | 2014 | 329 | 3561 | 8.6[6.8–10.7] | 0.5[0.3–0.8] | 94.2 |
| Chad | 2014 | 601 | 5724 | 50.6[46.0–55.2] | 17.4[15.2–19.7] | 65.6 |
| Egypt | 2014 | 881 | 14198 | 93.2[92.4–93.9] | 24.6[23.4–25.9] | 73.6 |
| Total | | 8396 | **93,063** | **60.0[58.9–61.1]** | **21.7[21.0–22.3]** | **63.8** |

All numbers and percentages were weighted.

sampling units (PSU). Then, households were selected from the clusters. The eligible (women aged 15 to 49 years old) members of the selected households were then interviewed. The surveys were conducted using similar protocols except for variations in data collected among countries. The surveys were conducted by trained interviewers using similarly structured questionnaires using standardized questions to assess women sexual and reproductive history, attitudes, knowledge and practices and relevant background characteristics.

## Data and measurement of variables

In this study, we analyzed data of 93,063 mother-daughter pair from 8,396 communities nested within 14 countries for the descriptive and the bivariable analysis (Fig 1).

**Outcome variable.**   The outcome variable is whether a mother had FGM for any of her daughters. All respondents were asked a series of questions including whether they had FGM or not. They were then asked questions if they have any living female child. Those who answered in affirmative were then asked questions on each of their living daughters to determine whether they had FGM for the daughters, age of the daughters at the time of FGM and how the cutting was carried out.

Women were classified into either they have ever had FGM for any daughter or not. For women that have at least two daughters, we first assessed the status of the youngest daughter. If the youngest daughter had FGM then the woman is classified to have had FGM for at least one daughter else, we then assessed the status of the next daughter until the oldest daughter. These steps were taken to avoid studying more than one daughter of the same woman. This will help to avoid correlations and dependencies in the characteristics of the mothers. We included only respondents (women) with response on at least one daughter's FGM status in this study. This was to allow mother-daughter analysis of FGM practice. Our analysis may, therefore, differ slightly from the published estimates on level of FGM among women and children by the DHS for each of the countries. Therefore, the outcome variable is a binary outcome: 1 = "Had FGM performed for at least one daughter", 0 = "Did not perform FGM for any daughter".

**Explanatory variables.**   The explanatory variables are categorized into three distinct levels: The levels are individual, neighbourhood and country levels, as shown in Fig 1.

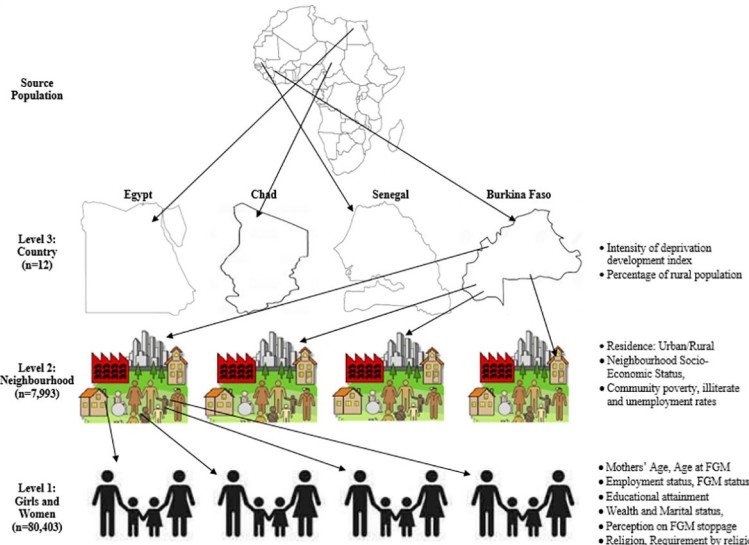

**Fig 1. The hierarchical nature of the data used in this study (Source: Authors drawing).**

*Individual-level factors*. The following individual-level factors about the mothers and their households were included in the analysis: mothers' current age (15 to 19, 20 to 24, 25 to 34 and 35 years or older), employment status (working or not working), education (no education, primary or secondary or higher), marital status (currently, formerly and never married), religion (Islam, Catholics, other Christianity, others). We separated the Catholics from other Christians women because they have slightly different doctrines on some sexual and reproductive issues [23]. Others are household wealth index (poorest, poorer, middle, richer and richest), mothers' FGM status (Had FGM or not), mothers' age at FGM (never, infancy, 1–5, 6–14 and >+15 years), FGM is required by religion (yes/no), should FGM be continued? (Continue, stop, depends).

*Neighbourhood-level factors*. The term "neighbourhood" was used to describe clustering within the same geographical living environment based on the PSUs within the DHS data. The neighbourhood-level factors included were the place of residence (rural or urban), community unemployment rate, illiteracy rate, poverty rate, and neighbourhood socio-economic status (SES). The neighbourhood SES was operationalized with a principal component analysis to aggregate the neighbourhood factors from the proportion of respondents with no formal education, unemployed, household wealth quintile (asset index below 20% poorest quintile). Standardized scores with mean 0 and standard deviation 1 were generated and categorized into 5: 1 (highest) to 5 (lowest).

*Country-level factors*. The country-level variables were extracted from the human index reports published and available in the United Nations database[27,28]. The variables are countries' percentage rural population[28] and the intensity of deprivation[27], both been different measures of human development index (HDI). The two factors were categorized into two (low and high) levels, as shown in Fig 1 and Table 2.

**Ethical consideration.** This was a secondary data analysis of DHS with approval from ICF Macro. The DHS usually sought for ethical approval in the individual country before data collection and from the Institutional Review Board (IRB) of ICF Macro at Fairfax, Virginia in the USA.

## Data analysis

Basic descriptive statistics were used to describe the data and the respondents' characteristics (Table 1) and the prevalence of mothers' and daughters' FGM (Table 2). The bivariate binary logistic regression was used to compare the odds of daughters' genital mutilation across the countries (Table 3). We also computed the prevalence of daughters' FGM among mothers that had FGM and those without FGM across the countries (Table 3). Finally, the binary multivariable multilevel logistic regression models were used to identify the contribution of the individual, community and country-level factors associated with the FGM among female children (Table 4). We used the binary multivariable model embedded in the MLWin 3.03 module in Stata statistical package version 16. The first order marginal quasi-likelihood linearization (MQL1) was adopted for the estimation algorithms using the iterative generalized least squares (IGLS). We applied sampling weights to the data, and statistical significance was determined at 5%.

However, Tanzania and Niger were not included in the multivariable analysis because the countries did not capture data on respondents' religion and if FGM was a religions' requirement or not. Respondents who did not give birth to any female child, or who has no living daughter or who were selected for male questionnaire according to the survey protocol were not asked questions on daughter's FGM. Of the 95,507 women that were asked questions on daughter's FGM, 93,063 (97,8%) provided valid responses (S1 Table). The remaining 2.2% with invalid responses on FGM among daughters were excluded from further analysis.

**Table 2. Pooled descriptive characteristics of study respondents.**

| Background characteristics | Frequency (%) | Prevalence of mother FGM | | Prevalence of daughter FGM | |
|---|---|---|---|---|---|
| | | % | 95% CI | % | 95% CI |
| **Mother Age (years)** | | | | | |
| 15–19 | 2930(3.1) | 57.9 | [55.4–60.4] | 9.0 | [7.6–10.5] |
| 20–24 | 12139(13) | 55.8 | [54.3–57.4] | 10.8 | [9.9–11.8] |
| 25–34 | 39636(42.6) | 58.3 | [57.1–59.5] | 17.0 | [16.3–17.8] |
| 35–49 | 38358(41.2) | 63.3 | [62.1–64.5] | 31.0 | [30.0–32.0] |
| **Age @ 1st Marriage** | | | | | |
| Never | 2653(2.9) | 41.5 | [38.6–44.5] | 5.6 | [4.5–6.8] |
| B4 age 15 | 14138(15.2) | 61.7 | [60.1–63.3] | 31.5 | [30.2–32.8] |
| 15–19 years | 48028(51.6) | 61.8 | [60.6–63.0] | 22.7 | [21.9–23.4] |
| 20+ years | 28244(30.3) | 57.9 | [56.5–59.3] | 16.5 | [15.8–17.3] |
| **Mother had FGM** | | | | | |
| No | 36675(39.4) | NA | NA | 3.1 | [2.8–3.5] |
| Yes | 56388(60.6) | NA | NA | 34.0 | [33.2–34.8] |
| **Mother Age at FGM (years)** | | | | | |
| < 1 (Infancy) | 15505(29.3) | NA | NA | 44.3 | [42.6–46.0] |
| 1–5 | 5021(9.5) | NA | NA | 39.8 | [37.6–42.0] |
| 6–14 | 28916(54.6) | NA | NA | 29.2 | [28.4–30.1] |
| ≥15 | 4422(6.6) | NA | NA | 21.0 | [19.2–22.9] |
| **Religion** | | | | | |
| Catholic | 7215(8.6) | 44.5 | [41.9–47.0] | 7.9 | [6.9–8.9] |
| Other Christians | 15943(19.0) | 42.0 | [40.0–43.9] | 10.1 | [9.2–11.1] |
| Islam | 57064(68.1) | 76.4 | [75.1–77.7] | 30.8 | [29.9–31.7] |
| Others | 3519(4.2) | 47.4 | [43.3–51.5] | 13.5 | [11.6–15.8] |
| **Highest educational level** | | | | | |
| No education | 50148(53.9) | 68.0 | [66.7–69.3] | 29.9 | [29.0–30.8] |
| Primary | 20294(21.8) | 41.6 | [40.1–43.1] | 11.3 | [10.6–12.0] |
| Secondary | 18522(19.9) | 61.1 | [59.6–62.7] | 14.9 | [14.1–15.7] |
| Higher | 4091(4.4) | 54.4 | [51.8–56.9] | 7.5 | [6.5–8.6] |
| **Marital status** | | | | | |
| Currently in union | 83863(90.1) | 61.4 | [60.2–62.5] | 22.4 | [21.7–23.1] |
| Formerly in union | 6547(7) | 49.9 | [47.9–51.9] | 18.6 | [17.3–19.8] |
| Never in union | 2653(2.9) | 41.5 | [38.6–44.5] | 5.6 | [4.5–6.8] |
| **Currently working** | | | | | |
| No | 55684(59.8) | 56.5 | [55.2–57.8] | 21.4 | [20.6–22.2] |
| Yes | 37379(40.2) | 65.4 | [64.0–66.8] | 22.1 | [21.2–22.9] |
| **Wealth quintiles** | | | | | |
| Poorest | 20398(21.9) | 65.4 | [63.6–67.1] | 28.7 | [27.5–30.0] |
| Poorer | 18260(19.6) | 63.3 | [61.6–64.9] | 24.8 | [23.7–25.9] |
| Middle | 18054(19.4) | 62.0 | [60.4–63.7] | 21.8 | [20.8–22.9] |
| Richer | 18648(20.0) | 58.5 | [56.7–60.2] | 19.5 | [18.5–20.6] |
| Richest | 17703(19.0) | 50.6 | [48.6–52.6] | 13.3 | [12.3–14.3] |
| **FGM required by religion** | | | | | |
| No | 57062(66.4) | 42.9 | [41.7–44.1] | 11.1 | [10.6–11.7] |
| Yes | 28858(33.6) | 91.8 | [91.1–92.4] | 43.1 | [42.0–44.2] |
| **FGM should be** | | | | | |
| Continued | 31779(35.5) | 93.8 | [93.2–94.3] | 47.3 | [46.2–48.4] |

**Table 2.** (Continued)

| Background characteristics | Frequency (%) | Prevalence of mother FGM | | Prevalence of daughter FGM | |
|---|---|---|---|---|---|
| | | % | 95% CI | % | 95% CI |
| Stopped | 54979(61.4) | 40.9 | [39.7–42.1] | 7.5 | [7.1–7.9] |
| Depends | 2823(3.2) | 53.1 | [49.9–56.3] | 24.0 | [22.0–26.2] |
| **Type of place of residence** | | | | | |
| Urban | 31618(34.0) | 52.5 | [50.7–54.4] | 15.9 | [14.9–16.9] |
| Rural | 61445(66.0) | 63.6 | [62.2–64.9] | 24.4 | [23.6–25.2] |
| **Community Socio-economic status** | | | | | |
| Highest | 18653(20.0) | 49.3 | [47.1–51.5] | 13.9 | [12.9–15.1] |
| 2 | 18627(20.0) | 57.4 | [54.8–60.0] | 19.9 | [18.5–21.5] |
| 3 | 18617(20.0) | 62.8 | [60.3–65.3] | 23.3 | [21.8–25.0] |
| 4 | 18577(20.0) | 63.7 | [61.0–66.4] | 23.7 | [22.2–25.3] |
| Lowest | 18589(20.0) | 66.8 | [64.4–69.2] | 27.5 | [26.0–29.1] |
| **Deprivation intensity** | | | | | |
| Low deprivation | 4970(5.3) | 44.8 | [40.8–48.8] | 13.1 | [11.3–15.2] |
| High deprivation | 88093(94.7) | 60.8 | [59.7–62.0] | 22.1 | [21.4–22.8] |
| **Rural percentage (%)** | | | | | |
| Low rural % | 21738(23.4) | 70.7 | [69.0–72.3] | 17.5 | [16.6–18.5] |
| High rural % | 71325(76.6) | 56.8 | [55.5–58.1] | 22.9 | [22.1–23.7] |
| Total | 93063(100) | 60.0 | [58.9–61.1] | 21.7 | [21.0–22.3] |

NA Not applicable All numbers and % were weighted.

**Binary multivariable hierarchical model.** In an attempt to arrive at the best model that best predict mothers' practice of FGM on their children, binary multilevel binary logistic regression models were fitted. We identified and fitted five models, and the results of these

**Table 3. The odds and distribution of daughters' FGM among mothers in Africa.**

| Country | Odds of Daughters' FGM Crude OR(95% CI) | Mothers with FGM | Mothers without FGM | Prevalence of Daughters' FGM among mothers that | | |
|---|---|---|---|---|---|---|
| | | | | Had FGM | Had no FGM | Overall |
| Nigeria | 63.1(42.1–94.5) | 2988 | 5303 | 55.6 | 16.6 | 31.2 |
| Kenya | 11.7(7.7–17.6) | 1904 | 5322 | 13.3 | 0.2 | 3.6 |
| Tanzania | 0.9(0.6–1.6) | 901 | 5011 | 3.3 | 0.03 | 0.5 |
| Senegal | 44.7(29.8–67) | 2203 | 5481 | 50.9 | 0.5 | 15.0 |
| Ethiopia | 61.2(40.7–92) | 3059 | 770 | 28.1 | 4.7 | 23.4 |
| Burkina Faso | 30.5(20.4–45.8) | 8260 | 1804 | 21.1 | 1.2 | 17.5 |
| Guinea | 152.4(101.7–228.5) | 5393 | 166 | 49.9 | 7.7 | 48.6 |
| Cote d'Ivoire | 27.5(18.3–41.5) | 2178 | 2686 | 28.4 | 0.7 | 13.1 |
| Mali | 398.7(264.7–600.5) | 292 | 218 | 83.1 | 14.6 | 78.3 |
| Niger | 5.2(3.3–8.1) | 162 | 3082 | 34.4 | 2.9 | 4.5 |
| Sierra Leone | 68.3(45.6–102.2) | 9423 | 336 | 32.3 | 3.6 | 31.4 |
| Togo | 1.0 (Reference) | 306 | 3255 | 5.3 | 0.1 | 0.5 |
| Chad | 37.1(24.7–55.8) | 2895 | 2829 | 33.7 | 0.6 | 17.4 |
| Egypt | 54.0(36.1–80.9) | 13234 | 964 | 26.3 | 1.1 | 24.6 |
| Total | | 55835 | 37228 | 34.0 | 3.1 | 21.7 |

All numbers were weighted.

**Table 4. Individual, neighbourhood and country level factors associated with daughters' FGM.**

| Characteristics | Null Model | | Model II | | Model III | | Model IV | | Model V | |
|---|---|---|---|---|---|---|---|---|---|---|
| | aOR(95% CI) | p-value | aOR(95% CI) | p-value | aOR(95% CI) | p-value | aOR(95% CI) | p-value | aOR(95% CI) | p-value |
| **Fixed Effect** | 0.28(0.16–0.49) | <0.001 | 0.00(0.00–0.00) | <0.001 | 0.18(0.1–0.31) | <0.001 | 0.19(0.05–0.79) | 0.022 | 0(0–0) | <0.001 |
| Constants | | | | | | | | | | |
| **Mothers Characteristics** | | | | | | | | | | |
| **Mothers' Current age(years)** | | | | | | | | | | |
| 15–19 | Reference | | | | | | | | | |
| 20–24 | | | 1.52(1.28–1.80) | <0.001 | | | | | 1.53(1.29–1.81) | <0.001 |
| 25–34 | | | 3.43(2.93–4.03) | <0.001 | | | | | 3.51(3.00–4.11) | <0.001 |
| > = 35 | | | 9.17(7.81–10.77) | <0.001 | | | | | 9.55(8.15–11.2) | <0.001 |
| **Mother education** | | | | | | | | | | |
| None | | | 2.43(2.10–2.80) | <0.001 | | | | | 2.53(2.16–2.96) | <0.001 |
| Primary | | | 1.92(1.66–2.22) | <0.001 | | | | | 2.01(1.72–2.36) | <0.001 |
| Secondary | | | 1.69(1.47–1.94) | <0.001 | | | | | 1.77(1.52–2.07) | <0.001 |
| Higher | Reference | | | | | | | | | |
| **Marital status** | | | | | | | | | | |
| currently married | Reference | | | | | | | | | |
| Formerly Married | | | 1.05(0.96–1.15) | 0.286 | | | | | 1.07(0.97–1.17) | 0.160 |
| Never Married | | | 0.57(0.46–0.69) | <0.001 | | | | | 0.59(0.48–0.72) | <0.001 |
| **Religion** | | | | | | | | | | |
| Catholic | Reference | | | | | | | | | |
| Other Christian | | | 1.00(0.88–1.14) | 0.963 | | | | | 0.99(0.87–1.13) | 0.926 |
| Islam | | | 1.98(1.77–2.21) | <0.001 | | | | | 1.94(1.73–2.17) | <0.001 |
| None | | | 1.35(1.14–1.60) | <0.001 | | | | | 1.32(1.12–1.57) | 0.001 |
| **Household wealth status** | | | | | | | | | | |
| Poorest | | | 1.6(1.47–1.74) | <0.001 | | | | | 1.36(1.22–1.51) | <0.001 |
| Poorer | | | 1.45(1.33–1.58) | <0.001 | | | | | 1.24(1.12–1.38) | <0.001 |
| Middle | | | 1.29(1.19–1.41) | <0.001 | | | | | 1.14(1.03–1.25) | 0.010 |
| Richer | | | 1.26(1.16–1.36) | <0.001 | | | | | 1.18(1.08–1.28) | <0.001 |
| Richest | Reference | | | | | | | | | |
| **Currently employed** | | | | | | | | | | |
| Yes | Reference | | | | | | | | | |
| No | | | 1.04(0.99–1.10) | 0.130 | | | | | 1.05(1.00–1.11) | 0.050 |
| **Mothers' FGM Status** | | | | | | | | | | |
| No FGM | Reference | | | | | | | | | |
| Had FGM | | | 13.95(4.85–40.17) | <0.001 | | | | | 14.67(4.82–44.69) | <0.001 |
| **Mothers' age @ FGM (years)** | | | | | | | | | | |
| Infancy | Reference | | | | | | | | | |
| 1–5 years | | | 0.99(0.91–1.08) | 0.869 | | | | | 0.99(0.91–1.08) | 0.883 |
| 6–14 | | | 0.76(0.71–0.81) | <0.001 | | | | | 0.77(0.72–0.82) | <0.001 |
| After 15 | | | 0.55(0.50–0.61) | <0.001 | | | | | 0.55(0.50–0.61) | <0.001 |
| Didn't | | | 1.51(0.52–4.33) | 0.448 | | | | | 1.50(0.49–4.57) | 0.472 |
| **Religion Required** | | | | | | | | | | |
| Yes | | | 1.46(1.38–1.54) | <0.001 | | | | | 1.47(1.39–1.55) | <0.001 |
| No | Reference | | | | | | | | | |
| **FGM should be** | | | | | | | | | | |
| stopped | Reference | | | | | | | | | |

(*Continued*)

**Table 4.** (Continued)

| Characteristics | Null Model | | Model II | | Model III | | Model IV | | Model V | |
|---|---|---|---|---|---|---|---|---|---|---|
| | aOR(95% CI) | p-value | aOR(95% CI) | p-value | aOR(95% CI) | p-value | aOR(95% CI) | p-value | aOR(95% CI) | p-value |
| Continued | | | 3.92(3.68–4.18) | <0.001 | | | | | 4.02(3.77–4.29) | <0.001 |
| Depends | | | 2.08(1.81–2.38) | <0.001 | | | | | 2.13(1.86–2.44) | <0.001 |
| **Neighbourhood characteristics** | | | | | | | | | | |
| Rural residence | | | | | 1.57(1.48–1.66) | <0.001 | | | 1.25(1.15–1.36) | <0.001 |
| High poverty rate | | | | | 1.09(1.03–1.15) | 0.003 | | | 1.08(1.01–1.16) | 0.018 |
| High illiteracy rate | | | | | 1.02(0.96–1.08) | 0.501 | | | 0.97(0.91–1.03) | 0.320 |
| High unemployment rate | | | | | 1.21(1.15–1.28) | <0.001 | | | 1.15(1.08–1.23) | <0.001 |
| **Country Characteristics** | | | | | | | | | | |
| **Rural Population percentage** | | | | | | | | | | |
| High | | | | | | | | | | |
| low | | | | | | | 0.52(0.06–4.19) | 0.539 | 0.71(0.16–3.06) | 0.646 |
| **Intensity of deprivation** | | | | | | | | | | |
| Low | | | | | | | | | | |
| High | | | | | | | 1.6(0.35–7.42) | 0.546 | 2.2(0.75–6.51) | 0.153 |
| **Random effects** | | | | | | | | | | |
| **Country-level** | 1.2(0.3–2.0) | | 0.6(0.1–1.1) | | 1.2(0.3–2.0) | | 1(0.3–1.8) | | 0.5(0.1–0.9) | |
| Variance (95 CI) | 21.6(6.7–32.2) | | 13.7(3.0–22.0) | | 21.9(6.8–32.5) | | 19.3(5.9–29.3) | | 11.4(2.5–18.7) | |
| VPC (%) | 2.8(1.7–3.9) | | 2.1(1.4–2.7) | | 2.8(1.7–3.9) | | 2.6(1.6–3.6) | | 2(1.3–2.5) | |
| MOR (%, 95% CI) | Reference | | 47.9(46.1–60.7) | | 0.4(0.0–0.9) | | 11.2(11.1–11.7) | | 57.3(55.7–68.0) | |
| Explained variation (%) | | | | | | | | | | |
| **Neighbourhood-level** | | | | | | | | | | |
| Variance (95 CI) | 0.9(0.9–1.0) | | 0.5(0.5–0.6) | | 0.9(0.8–0.9) | | 1.0(1.0–1.1) | | 0.6(0.5–0.6) | |
| VPC (%) | 38.8(26.3–47.6) | | 25.4(15.2–33.4) | | 38.3(25.6–47.3) | | 38.2(27.1–46.4) | | 24.1(15.5–31.2) | |
| MOR (%, 95% CI) | 2.5(2.4–2.6) | | 2.0(1.9–2.0) | | 2.4(2.4–2.5) | | 2.6(2.5–2.7) | | 2.0(2.0–2.1) | |
| Explained variation (%) | Reference | | 44.0(42.2–46.1) | | 5.1(4.8–5.1) | | 8.9(8.9–9.0) | | 40.2(38.2–41.8) | |
| **Model fit statistics** | | | | | | | | | | |
| Deviance (-2LL) | 12134.56 | | 10234.88 | | 12098.71 | | 12101.31 | | 10141.98 | |
| **Sample Size** | | | | | | | | | | |
| Individual | 83758 | | 73353 | | 83758 | | 83758 | | 73353 | |
| Country | 12 | | 12 | | 12 | | 12 | | 12 | |
| Neighbourhood | 7817 | | 7778 | | 7817 | | 7817 | | 7778 | |

AOR Adjusted odds ratio, CI confidence interval, MOR median odds ratio, VPC variance partition coefficient. Tanzania has no records or religion and was excluded from the analysis. **Had all independent variables in the model.

models were reported in Table 3. The first model was the null model (Model I) to assess the variation due to the neighbourhood- and the country-specific random effects without any explanatory variable. Model II had only the individual-level variables conditional on the neighbourhood and country-specific random effects while Model III had only the neighbourhood level variables conditional on the neighbourhood and country-specific random effects. In

contrast, Model IV examined the country-level variables conditional on the neighbourhood and country-specific random effects. Finally, we developed Model V to estimate the odds of individual, neighbourhood and country-level factors conditional on the neighbourhood and country-specific random effects.

Typically, the multilevel analysis provides estimates for both fixed and random effects. The fixed effects are the lowest hierarchy in the multilevel model (individual mothers' characteristics in this study) while the random effects are based on the contribution of the upper levels (higher hierarchies), these are the neighbourhood and country levels in this study. A three-level variance model for $\pi_{ijk} = P(y_{ijk} = 1)$ as shown in Eqs (1) and (2) was determined.

$$y_{ijk} \sim Binomial \left( n_{ijk}, \pi_{ijk} \right) \tag{1}$$

In the Binomial function, there are two possible possibilities: had FGM for at least one daughter or not; $y_{ijk}$ is the daughter $i$ of neighbourhood $j$ from country $k$, while the probability that the mother of daughter $i$ of neighbourhood $j$ from country $k$ had FGM for at least a daughter is denoted by $\pi_{ijk}$.

The fitted models were based on the hierarchical logistic regression model with mixed outcomes consisting of the fixed and random parts, as shown in Eq (2).

$$log \left( \frac{\pi_{ijk}}{1 - \pi_{ijk}} \right) = logit \left( \pi_{ijk} \right) = \underbrace{\beta_0 + \sum_{p=1}^{P} \beta_p X_{pijk}}_{Fixed} + \underbrace{U_{0jk} + V_{0k}}_{Random} + e_{ijk} \tag{2}$$

Where $\frac{\pi_{ijk}}{1-\pi_{ijk}}$ is the odds that $y_{ijk} = 1$, $\beta_0$ is the fixed intercept, $\beta_p$ are the regression coefficients of the covariates $X_p$, $U_{ojk}$ is the random effect of daughters in the neighbourhood $j$ in the country $k$, and $U_{ojk}$ is the random effect of country $k$, $e_{ijk}$ is the noise such that

$$e_{ijk} \sim N(0, \sigma_e^2),$$

$$U_{ojk} \sim N(0, \sigma_U^2) \text{ and}$$

$$V_{ok} \sim N(0, \sigma_V^2).$$

Other statistical details of the model have been published earlier [29–31]. Note that $p(\beta_0) \propto 1$, $p(\sigma_U^2) \sim \Gamma^{-1}(\varepsilon, \varepsilon)$ and $p(\sigma_V^2) \sim \Gamma^{-1}(\varepsilon, \varepsilon)$. We assumed diffuse priors and used an improper uniform prior for $\beta_0$ and a commonly used conjugate inverse Gamma prior for $\sigma_U^2$ and $\sigma_V^2$. We implemented the regression Eq (3) in the MLWin 3.03 module embedded in Stata statistical package version 16 with the individuals ($i's$) as level 1, the neighbourhoods ($j's$) as level 2 and countries ($k's$) as level 3.

*Fixed effects.* The primary outcomes of all the five models were the measures of the association expressed as odds ratios (ORs) with their 95% confidence intervals (CI). It is an expression of the likelihood of the outcome variable in the categories of a variable compared to the reference category of that variable.

*Random effects.* This is a measure of variation that is explained by the higher levels of the hierarchies in the data. The measures of variations were explored using the variance partition coefficient (VPC) and the median odds ratio (MOR). We reported the random effects in terms of the odds using the methods proposed by Larsen et al. on neighbourhood effects [32]. The VPC is a summary of the degree of clustering in the data and it is a reasonable interpretation of the Intra-class correlation (ICC) which measures the extent to which the $y_{ijk}'s$ in the same

neighbourhoods and countries resemble each other as compared to those from other clusters [33]. Therefore, the VPC was used to measure the proportion of the total variance which are accounted for at the neighbourhood and the country levels and computed as $\sigma_U^2/(\sigma_U^2 + \sigma_V^2 + \sigma_e^2)$ and $\sigma_V^2/(\sigma_U^2 + \sigma_V^2 + \sigma_e^2)$ respectively.

The MORs are the measures of the variance of the odds ratio in higher levels (neighbourhood or country), and it estimates the probability of daughters' FGM that can be attributed to any of the neighbourhood and country factors. A MOR of 1 is an indication that there is no neighbourhood or country variability. A MOR > 1 suggests that the contextual effects -neighbourhood/country variability- is significant. A higher MOR indicates that the contextual effects for understanding the probability of a mother to have FGM performed for her daughter is higher. This statistical analytic approach has been used and reported in the literature [29–31,34,35].

**Model fit.** Multicollinearity within explanatory variables was assessed by examining the variance inflation factor (VIF) [36]. There was no evidence of multicollinearity. We used the model deviance computed from the -2loglikelihood to assess and identify the best model that fitted the data. Lower deviance indicates a better model fit.

## Results

The details of countries, year of data collection, distribution of women by countries, and the prevalence of FGM among mothers and daughters were reported in Table 1. A total of 93,063 women age 15 to 49 years (Level 1) in 8,396 neighbourhoods (Level 2) from 14 African countries. The overall prevalence of FGM among mothers and their daughters was 60% and 21.7%, respectively, corresponding to 63.8% reduction in the mother-daughter ratio of FGM. As shown in Table 1 and Fig 2A, the prevalence of FGM varied widely across countries; the prevalence of FGM among mothers ranged from 8.6% in Togo and to over 90.0% in Mali, Egypt, Guinea and Sierra Leone. The prevalence of FGM among daughters in Togo and Tanzania were less than one percent, 48.6% in Guinea, with the highest prevalence of 78.3% reported in Mali (Table 1 and Fig 2B).

The percentage reduction in mother-daughter FGM prevalence was highest in Tanzania (96.7%) and Togo (94.2%), while only 10.0%, 13.6% and 15.9% reduction observed in Niger, Nigeria and Mali respectively (Table 1 and Fig 3). The reported age when FGM was performed among mothers and daughters ranged from 0–20 years or older. One-third of mothers had FGM at infancy and almost 2 in 5 of daughters had FGM at infancy. The mothers' mean age at FGM was 3.4 years (SD = 3.2), Median = 2.3 years, Range = 0 to 15 years. While over half of the daughters had FGM by their second birthday, only one-third of mothers had FGM at the same age (Fig 4 and S2 Table).

The descriptive statistics of the individual, community and country-level characteristics with their corresponding prevalence of mothers and their daughters' FGM were reported in Table 2. The proportion of mothers were evenly distributed across wealth quintiles, only 3.1% of mothers were adolescent (< 20 years); most (90.1%) of the mothers were currently in a union, and only 2 in 5 (40.2%) mothers were currently working. About half (53.9%) of the mothers had no formal education and 68.1% were Muslims. More than half of the mothers reported that they had FGM between ages 6 and 14 years and about 3 in 10 of mothers reported that FGM was required by their religion and also believed that FGM should continue.

We presented the distribution of mother and daughter's background characteristics and prevalence of FGM among daughters as shown in Table 2. The prevalence of daughters' FGM was lower among adolescent mothers compared with older mothers (9.0% vs 31.0%); just as a higher prevalence of mothers' FGM was found among women aged 35 years or older (57.9%

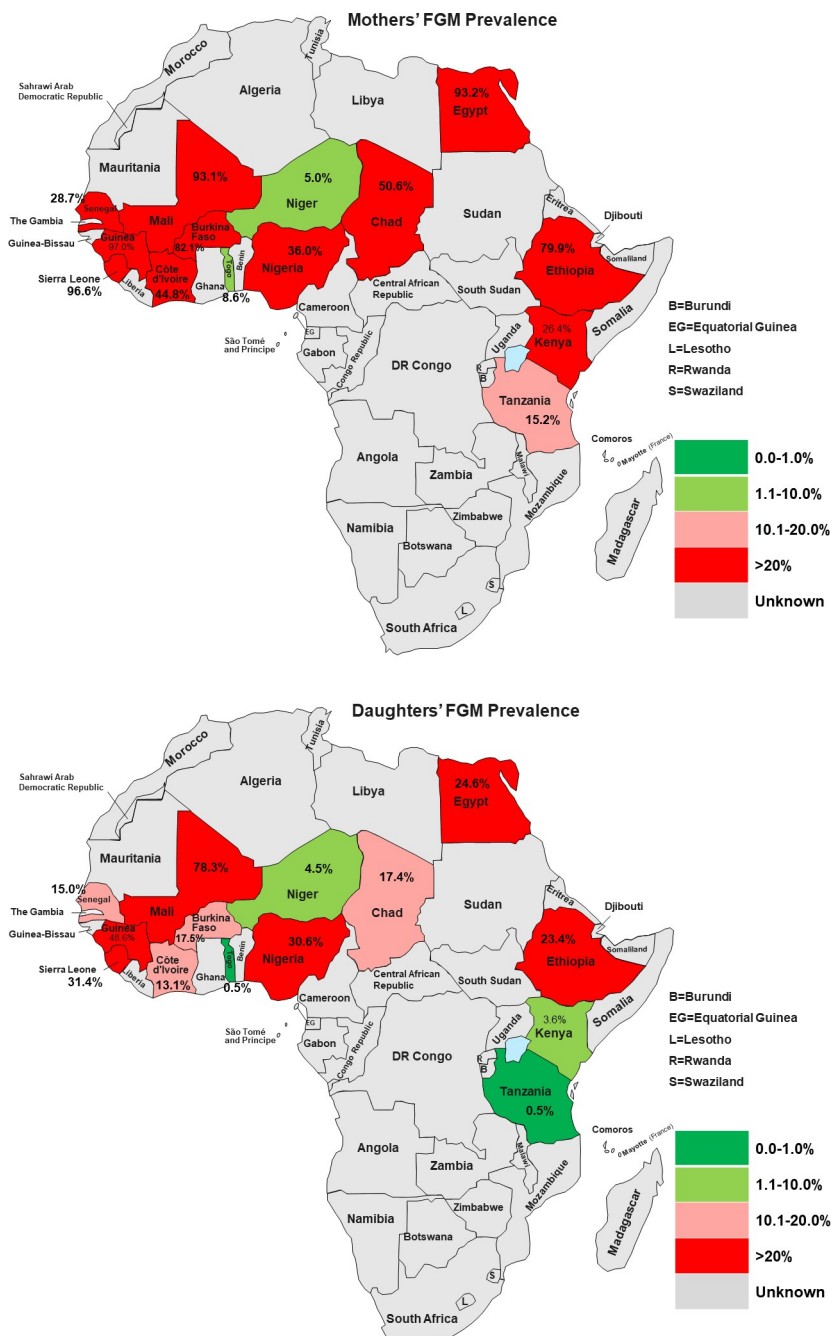

**Fig 2.** a: Prevalence of Mothers' FGM in African Countries (Source: Authors drawing). b: Prevalence of Daughters' FGM in African Countries (Source: Authors drawing).

vs 63.3%). Mothers who reported having FGM had a high FGM prevalence of 34.0% among daughters compared to 3.1% among mothers who had no FGM. Similarly, mothers whose FGM was performed at infancy or before the fifth birthday had FGM prevalence greater than 40% among daughters compared with 29.2% and 21.0% among mothers who had FGM between age 6 and 14 years and at 15 years or older age respectively.

The prevalence of daughters' FGM was highest among Muslim mothers (30.8%) than those of other religions; mothers who reported that their religion required FGM had daughters'

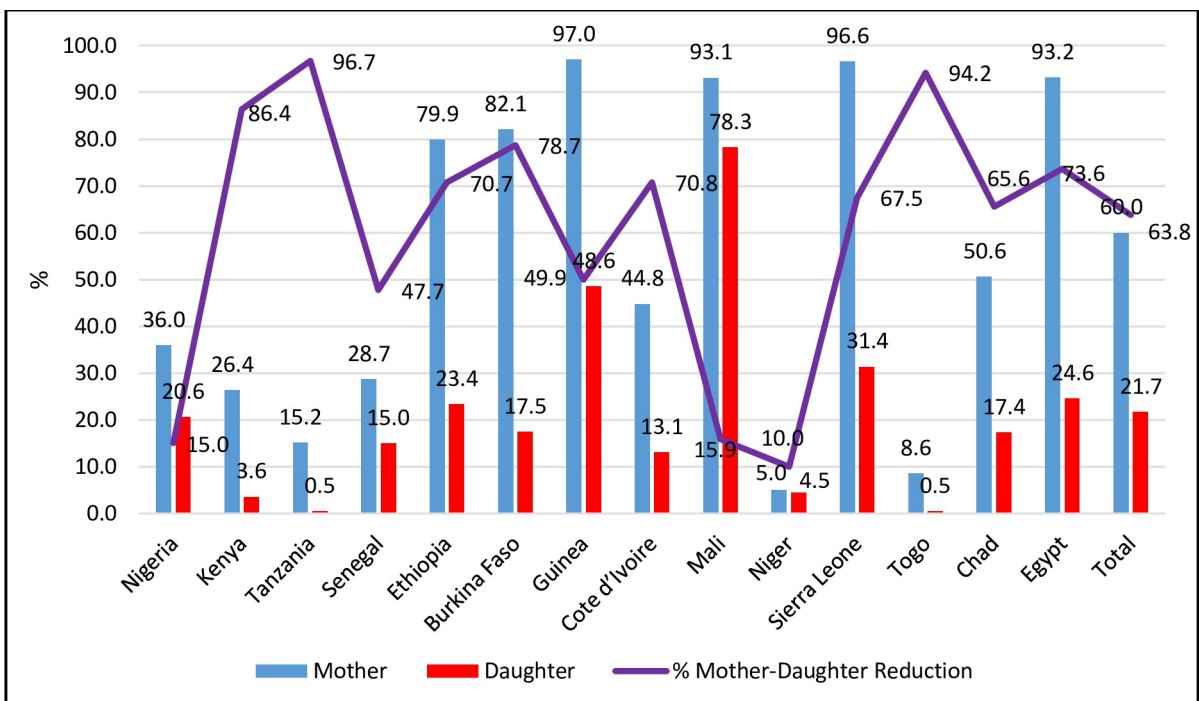

**Fig 3. Comparison of the prevalence of FGM among mothers and daughters and the percentage reduction.**

FGM prevalence of 43.1% compared with 11.1% among those whose religion did not require FGM. Although, an inverse relationship was observed between mothers' educational level and prevalence of daughters' FGM; the prevalence of daughter's FGM was higher among women with secondary education (14.9%) compared with 29.9% among women with no education. The prevalence of daughters' FGM among mothers who believed that FGM should be continued was 47.3% compared with 7.5% among those who want FGM stopped. The prevalence of FGM among daughters whose mothers reside in rural areas (24.4%) were higher than mothers in the urban setting (15.9%). Higher household wealth quintile was associated with the lowest prevalence of FGM among daughters. Countries with high deprivation intensity and high rural population percentage had a higher prevalence of FGM among daughters compared with countries that have low deprivation intensity and low rural population percentage.

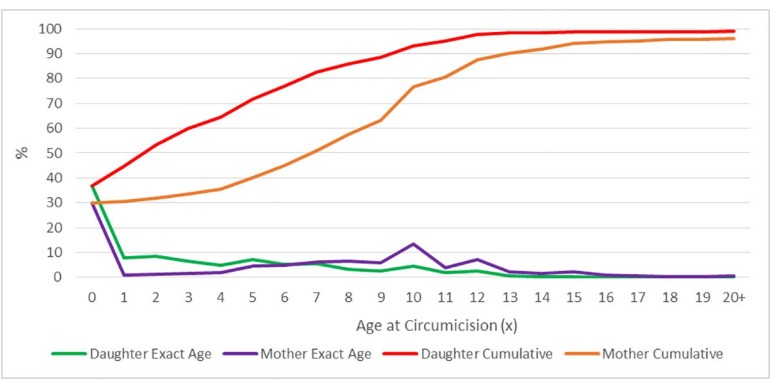

**Fig 4. Comparison of mothers' and daughters' age at FGM.**

## The odds of procurement and uptake of daughters' FGM among mothers with and without FGM across the countries

The crude odds of daughters' FGM revealed that Nigeria, Kenya, Senegal, Ethiopia, Burkina Faso, Guinea, Cote d'Ivoire, Mali, Sierra Leone, Niger, Chad, Egypt except Tanzania had a higher odds of daughters' FGM compared to Togo. The odds were about 400 times higher in Mali compared with Togo (Table 3). In all, 34% of mothers with FGM perpetuated the act among their daughters while 3% practised FGM on their daughters despite not having FGM themselves. The prevalence of daughters' FGM differed significantly across the countries. The procurement of FGM for daughters among mothers that were cut was highest in Mali at 83%, 56% in Nigeria and 3% in Tanzania while uptake of FGM for daughters of mothers without FGM was 17% in Nigeria, 15% in Mali and 0.03% in Tanzania (Table 3).

## Hierarchical analysis of factors associated with daughters' FGM

In all the models, the MOR and its confidence intervals were greater than 1 for both the neighbourhood and country random effects, which indicated that there exist significant variations among the neighbourhoods and countries in the probability of procuring FGM for their daughters. A comparison of the deviance statistics of each model revealed that Model V has the best performance as it provided the best prediction of factors associated with daughters FGM (Table 4). In Model V, the MOR for country effect was 2.0 (95% CI: 1.3–2.5) while the MOR for neighbourhood effect was 2.0 (95% CI: 2.0–2.1). Besides, Model V has the combined highest variation explained by the country-level and neighbourhood-level factors. Over 40% (95% CI: 38.2–41.8) of the variations in the odds of daughters' FGM were explained by neighbourhood effects while 57% (95% CI: 55.7–68.0) were explained by country effects.

The fitted model suggests that mothers aged 20 years or older compared to adolescent mothers, mothers who had a secondary or lower education compared to post-secondary, mothers in a lower household wealth quintile compared to the richest wealth quintile, and mothers who had FGM at age 6 years or older were more likely to perform FGM on their daughter. Also, mothers who reside in a rural neighbourhood (OR = 1.25, 95% CI: 1.15–1.36), with high poverty rate (OR = 1.08, 95% CI: 1.01–1.16) and unemployment rate (OR = 1.15, 95% CI: 1.08–1.23) had a higher odds of procuring FGM for their daughters.

## Discussion

The study provided the most extensive and up to date data in Africa on the practice, procurement, perpetuation, and initiation of daughters' FGM by mothers. The study showed that a large proportion of women in Africa have procured FGM for their daughters. Generally, one in five daughters were reported to have had FGM. It is of great concern that some women who never had FGM initiated FGM for their daughters. While the overall prevalence of daughters' FGM among women without FGM was 3.2%, it was as high as 17% in Nigeria and 15% in Mali. The reported overall prevalence of daughter's FGM was ten times higher among mothers that had FGM relative to those with no FGM. The prevalence of FGM among daughters varied by countries with the highest in Mali, followed by Guinea, Nigeria and Egypt. There were higher odds of daughters' FGM in other countries relative to Togo. However, the reduction in the percentage of mothers' FGM to daughters' FGM rate was lowest in Nigeria and Mali, and highest in Tanzania and Togo relative to other countries. The factors that were significantly associated with the risk of mothers having FGM for their daughters include individual maternal factors–mothers that were 20 years and above in age, had secondary or lower educational status, Islamic religion, from poorer or poorest household wealth quintiles and living in rural

neighbourhood community. Other associated risk factors of FGM in daughters were history of FGM in mothers, mothers whose FGM was performed at 6 years and above, and have a positive disposition towards the continuation of FGM practise. Living in rural residence and high unemployment environments are the two neighbourhood factors that were found to be associated with the report of FGM among daughters.

The findings from this study showed that the burden of FGM in Africa, especially among young children is high. Considering the differences in the current age of the daughters and their mothers, the rate of FGM among the young girls of today may surpass the rates among the current mothers by the time the girls complete their reproductive circle, particularly in Nigeria and Mali. Moreover, it is worrisome that this cultural practice is being continuously practised across different countries in daughters of women interviewed in the surveys despite the enactment of laws against the practice and colossal investment on prevention programmes [26,37]. The general submissive attitude of women towards cultural practice even when such culture is harmful has been widely reported in Africa. For example, women are often forced to accept some cultural rites during marriages or following the death of their spouse without any consequences [38,39]. These women are expected to transmit such cultures to their daughters. It is not also surprising that there were higher reports of FGM among daughters of women that suffered FGM compared to those whose mothers did not have FGM in this study. Besides, mothers that had their FGM at a very young age tend also pass the practice of FGM at infancy to their daughters. A plausible explanation is that the performance of FGM at infancy or early childhood in such communities is viewed as part of cultural childbirth rites [19].

The perpetuation of FGM among daughters of mothers that were not educated, and from lower socio-economic class as well as living in areas with high unemployment rates attest to the need to empower women with education and means of livelihood to reduce the risk of genital mutilation in the future. Educated and well-empowered women usually have a voice towards key decisions in the family, especially on their health and that of their children [37]. However, women's socio-economic status may be an influencer of FGM. A recent finding by Morhason-Bello et al. showed that women of high wealth quintiles were more likely to seek FGM from health practitioners compared to non-medical workers [23]. Similar to previous studies, religion was another significant factor that is associated with the possibility of a daughter to have FGM in this study [17,19]. Although the evidence is mixed among Islamic scholars towards FGM as part of a religious rite for women; some scholars believed that FGM should be performed for girls or new converts to Islam, but others opposed this practice in its entirety [17]. Despite this controversy, studies have consistently reported that high prevalence and risk of reporting among Muslim girls and women relative to those practising Christianity and other religions [19,26].

Apart from the individual risk factors, environment/community in which people resides might also influence attitude and risk towards FGM practice. This study clearly shows that daughters whose mothers reside in a rural community where cultural values are better enforced and in high unemployment environment are more predisposed to FGM. This finding shows that it is equally important to consider the environment of people when a policy or prevention programme on FGM is being planned in Africa to achieve a robust and optimal positive effect to reduce the practice. In order to stop FGM in Africa, potential mothers irrespective of their previous experience of FGM will need to be supported through aggressive community engagement to mitigate future continuation of this cultural practice.

## Strength and limitations

The findings from this study should be interpreted with caution. The analysis used secondary data that were collected between 2010 and 2018 in 14 countries to explain the procurement of

FGM for girls in Africa. The cross-sectional design of the data did not allow us to draw a causal association between the identified risk factors and the outcome variables. The difference in the time of data collection and different culture, as well as the specific political climate in each of the selected country, could potentially limit generalisability. For example, the last DHS for Burkina Faso and Cote d'Ivoire was 2010 and 2012, respectively, compared to Nigeria and Guinea that were conducted in 2018. Notwithstanding these limitations, this study used multi-level analysis to investigate the role of the individual, neighbourhood and country-level factors to assess the burden of FGM among future generations of women in Africa. The analysis also provided insight into the possible country-level specific differences in the procurement of FGM for daughters among the current generation of women.

## Conclusions

This study showed that the practice and procurement, hence the perpetuation of FGM among daughters of the present generation of women in Africa is common, particularly, among those from low social and educational status. This habit was commonly reported in a poor and rural environment with variation in the prevalence across countries in Africa. Interestingly, Niger has less than 5% prevalence of both mothers' and daughters' FGM but at 10%, the percentage mother-daughter reduction is high. While countries such as Togo and Tanzania have remarkable achievement in stopping FGM among both the mothers and their daughters, the burden of FGM remained high in countries such as Nigeria and Mali. The future burden of FGM in these two countries will be high. It is advocated that context-specific longitudinal data are collected to fully understand the role of each of the risk factors that aid the perpetuation of FGM in the continent in order to design sustainable interventions towards the elimination of FGM.

## Supporting information

**S1 Table. Distribution of the sampled women, proportion with living children, proportion asked question on daughters' FGM and valid responses by country.**
(DOCX)

**S2 Table. Comparison of mothers' and daughters' age at FGM.**
(DOCX)

## Acknowledgments

The authors acknowledge ICF for granting access and use of the data. We also acknowledge the owner of Geocurrent Maps (http://www.geocurrents.info/gc-maps) for allowing the use of their editable maps in Figs 1 and 2.

## Author Contributions

**Conceptualization:** Adeniyi Francis Fagbamigbe, Imran Oludare Morhason-Bello, Yusuf Olushola Kareem.

**Data curation:** Adeniyi Francis Fagbamigbe.

**Methodology:** Adeniyi Francis Fagbamigbe.

**Resources:** Imran Oludare Morhason-Bello.

**Visualization:** Adeniyi Francis Fagbamigbe.

**Writing – original draft:** Adeniyi Francis Fagbamigbe, Imran Oludare Morhason-Bello, Yusuf Olushola Kareem, Erhabor Sunday Idemudia.

**Writing – review & editing:** Adeniyi Francis Fagbamigbe, Imran Oludare Morhason-Bello, Yusuf Olushola Kareem, Erhabor Sunday Idemudia.

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
