## [Decision Letter · Decision Letter 0]

2 Sep 2020

PONE-D-20-19500

Perpetration of female genital mutilation in the next generation of women from 12 African countries: Evidence from hierarchical modelling of their attributes

PLOS ONE

Dear Dr. Fagbamigbe,

Thank you for submitting your manuscript to PLOS ONE. After careful consideration, we feel that it has merit but does not fully meet PLOS ONE’s publication criteria as it currently stands. Therefore, we invite you to submit a revised version of the manuscript that addresses the points raised during the review process.

We look forward to receiving your revised manuscript.

Kind regards,

Joshua Amo-Adjei, Ph.D

Academic Editor

PLOS ONE

Journal Requirements:

3. We note that Figures 1 and 2 in your submission contain map images which may be copyrighted.

a. You may seek permission from the original copyright holder of Figures 1 and 2 to publish the content specifically under the CC BY 4.0 license. 

4. We note that Figure 1 includes images of individuals.

As per the PLOS ONE policy (http://journals.plos.org/plosone/s/submission-guidelines#loc-human-subjects-research) on papers that include identifying, or potentially identifying, information, the individual(s) or parent(s)/guardian(s) must be informed of the terms of the PLOS open-access (CC-BY) license and provide specific permission for publication of these details under the terms of this license.

Please download the Consent Form for Publication in a PLOS Journal (http://journals.plos.org/plosone/s/file?id=8ce6/plos-consent-form-english.pdf).

The signed consent form should not be submitted with the manuscript, but should be securely filed in the individual's case notes. Please amend the methods section and ethics statement of the manuscript to explicitly state that the patient/participant has provided consent for publication: “The individual in this manuscript has given written informed consent (as outlined in PLOS consent form) to publish these case details”.

If you are unable to obtain consent from the subject of the photograph, you will need to remove the figure and any other textual identifying information or case descriptions for these individuals.

5. Please ensure that language is neutral and factual throughout the manuscript. We would for instance ask that you consider whether the term 'perpetration' in the title can be changed to a more neutral term.

6. Please amend either the abstract on the online submission form (via Edit Submission) or the abstract in the manuscript so that they are identical.

7. Please ensure that you refer to Figure 4 in your text as, if accepted, production will need this reference to link the reader to the figure.

8. Please include captions for your Supporting Information files at the end of your manuscript, and update any in-text citations to match accordingly. Please see our Supporting Information guidelines for more information: http://journals.plos.org/plosone/s/supporting-information

Reviewers' comments:

Reviewer's Responses to Questions

**Comments to the Author**

1. Is the manuscript technically sound, and do the data support the conclusions?

Reviewer #1: Yes

Reviewer #2: Yes

2. Has the statistical analysis been performed appropriately and rigorously? 

Reviewer #1: I Don't Know

Reviewer #2: Yes

3. Have the authors made all data underlying the findings in their manuscript fully available?

Reviewer #1: Yes

Reviewer #2: Yes

4. Is the manuscript presented in an intelligible fashion and written in standard English?

Reviewer #1: Yes

Reviewer #2: Yes

5. Review Comments to the Author

Reviewer #1: I found the manuscript well written, informative and touches on a critical problem bewildering most of the developing countries. Though it is academic paper, it will really benefit policy /decision makers who are willing to act on this historical injustice to women and girls. However, in section 2.3, line 119, the author mentioned that they will collect information or seek to understand on "how the FMG was done". maybe the author can recheck as it doesn't come out clearly in the results section. On the other note, it would be interesting to understand why Catholics were looked at separately from other Christians. There seems to be no mention the role of Catholics in literature review.

Reviewer #2: This is a well-carried out study that seeks to fill an important gap in reproductive literature in Africa. The paper was well written from background to conclusion. However, I have some few methodological suggestions which can help strengthen the paper.

1. Under line 114, the authors state “In this study, we analyzed data of 80403 girls from 7993 communities nested within 12 countries”. The authors should be clear whether the unit of analysis is ‘women with daughters’ or ‘girls’.

2. Under line 167-168, the authors state “Data were weighted and statistical significance determined at 5%.”. Can the authors indicate how data was weighted?

3. Can the authors indicate how they accounted for the complex sampling design of the surveys?

4. The authors pooled data from 12 African countries. Where is weighting? I do not mean the DHS weighting, but I mean your data weighting after pooling as you deal with multiple countries with wide variation in their population. You can refer to the reference below to understand what I mean.

Marriott, B. M., Campbell, L., Hirsch, E., & Wilson, D. (2007). Preliminary data from demographic and health surveys on infant feeding in 20 developing countries. The Journal of nutrition, 137(2), 518S-523S.

5. Under line 170-172, the authors state “Respondents with no female child and respondents with no information on whether the daughter was circumcised or not were excluded from further analysis.” This is an indication that there was missing data in each country’s dataset, can the authors provide the following information for each country

A. total women interviewed

B. Sample size by design

C. Selected women sample (this is already in Table 1).

D. Percentage with complete information on the inclusion criteria

E. Percentage of missing data.

6. Finally, can the authors explain how they treated missing data?

6. PLOS authors have the option to publish the peer review history of their article (what does this mean?). If published, this will include your full peer review and any attached files.

Reviewer #1: **Yes: **Patrick O. Okoth

Reviewer #2: **Yes: **Bright Opoku Ahinkorah

---

## [Author Response · Author response to Decision Letter 0]

1 Oct 2020

The new WHO approved name for genital mutilation is FGM since 2018/2019 and not FGM/C as suggested by the editorial office. FGM/C is now obsolete

September 13th 2020

Dear Editor

PLOS ONE 

Re: PONE-D-20-19500

Perpetration of female genital mutilation in the next generation of women from 12 African countries: Evidence from hierarchical modelling of their attributes

We the authors of above mentioned paper appreciate the efforts and comments of the editor and the eminent reviewers. We have addressed all these comments. A point-by-point response to the issues in our revised manuscript is listed below.

Please note that we used the file with the tracked changes to describe where the changes were made. 

Editor Comments:

Journal Requirements:

 THANK YOU. WE HAVE FORMATTED THE ENTIRE MANUSCRIPT TO MEET PLOS ONE STYLES

 THANK YOU. WE HAVE RNGAGED THE SERVICES OF NATIVE ENGLISH EDITOR IN THE COPYEDITTING

3. We note that Figures 1 and 2 in your submission contain map images which may be copyrighted.

 a. You may seek permission from the original copyright holder of Figures 1 and 2 to publish the content specifically under the CC BY 4.0 license. 

THE GEOGRAPHICAL MAPS AND OBJECTS USED IN THE FIGURES WERE NOT COPYRIGHTED. HOWEVER, WE HAVE OBTAINED A WRITTEN PERMISSION FROM THE OWNER (SEE THE ATTACHMENT IN THE “OTHERS”). THE TWO FIGURES WERE DRAWN BY THE AUTHORS, THE TEXTS IN THE FIGURES ARE THE WORKS OF THE AUTHORS. 

4. We note that Figure 1 includes images of individuals.

As per the PLOS ONE policy (http://journals.plos.org/plosone/s/submission-guidelines#loc-human-subjects-research) on papers that include identifying, or potentially identifying, information, the individual(s) or parent(s)/guardian(s) must be informed of the terms of the PLOS open-access (CC-BY) license and provide specific permission for publication of these details under the terms of this license.

Please download the Consent Form for Publication in a PLOS Journal (http://journals.plos.org/plosone/s/file?id=8ce6/plos-consent-form-english.pdf).

The signed consent form should not be submitted with the manuscript, but should be securely filed in the individual's case notes. Please amend the methods section and ethics statement of the manuscript to explicitly state that the patient/participant has provided consent for publication: “The individual in this manuscript has given written informed consent (as outlined in PLOS consent form) to publish these case details”.

If you are unable to obtain consent from the subject of the photograph, you will need to remove the figure and any other textual identifying information or case descriptions for these individuals.

 THANK YOU. WE HAVE REPLACED THE HUMAN IMAGES WITH NON-COPYRIGHTED OBJECTS

5. Please ensure that language is neutral and factual throughout the manuscript. We would for instance ask that you consider whether the term 'perpetration' in the title can be changed to a more neutral term.

 THANK YOU. WE AGREE WITH YOU AND CONSIDERED “PROCUREMENT PERPETUATION” TO BE MORE NEUTRAL THAN PERPETRATION

6. Please amend either the abstract on the online submission form (via Edit Submission) or the abstract in the manuscript so that they are identical.

THANK YOU. THE ABSTRACTS ARE NOW THESAME

7. Please ensure that you refer to Figure 4 in your text as, if accepted, production will need this reference to link the reader to the figure.

FIGURE 4 HAVE BEEN CITED

8. Please include captions for your Supporting Information files at the end of your manuscript, and update any in-text citations to match accordingly. Please see our Supporting Information guidelines for more information: http://journals.plos.org/plosone/s/supporting-information

THANK YOU. WE HAVE PROVIDED THE CAPTIONS FOR THE SUPPORTING INFORMATION FILES AT THE END OF YOUR MANUSCRIPT…….

Reviewers' comments:

Reviewer's Responses to Questions

Comments to the Author

1. Is the manuscript technically sound, and do the data support the conclusions?

Reviewer #1: Yes

Reviewer #2: Yes

 2. Has the statistical analysis been performed appropriately and rigorously?

 Reviewer #1: I Don't Know

Reviewer #2: Yes

 3. Have the authors made all data underlying the findings in their manuscript fully available?

 Reviewer #1: Yes

Reviewer #2: Yes

 4. Is the manuscript presented in an intelligible fashion and written in standard English?

 Reviewer #1: Yes

Reviewer #2: Yes

 5. Review Comments to the Author

Reviewer #1: I found the manuscript well written, informative and touches on a critical problem bewildering most of the developing countries. Though it is academic paper, it will really benefit policy /decision makers who are willing to act on this historical injustice to women and girls.

THANK YOU, WE HOPE TO DEVELOP A POLICY BRIEF FROM THIS ACADEMIC PAPER

However, in section 2.3, line 119, the author mentioned that they will collect information or seek to understand on "how the FMG was done". maybe the author can recheck as it doesn't come out clearly in the results section. 

THANK YOU, THIS WAS AN ERROR, CHECKING HOW FGM WERE DONE WAS OUTSIDE THE SCOPE OF THIS STUDY. WE HAVE REMOVED THE PHRASE.

On the other note, it would be interesting to understand why Catholics were looked at separately from other Christians. There seems to be no mention the role of Catholics in literature review.

WE SEPARATED THE CATHOLICS FROM THE OTHER CHRISTIANS BECAUSE LITERATURE SUGGESTED THEY HAVE SLIGHTLY DIFFERENT DOCTRINES ON SOME SEXUAL AND REPRODUCTIVE ISSUES

Reviewer #2: This is a well-carried out study that seeks to fill an important gap in reproductive literature in Africa. The paper was well written from background to conclusion. 

THANK YOU

However, I have some few methodological suggestions which can help strengthen the paper.

1. Under line 114, the authors state “In this study, we analyzed data of 80403 girls from 7993 communities nested within 12 countries”. The authors should be clear whether the unit of analysis is ‘women with daughters’ or ‘girls’.

THANK YOU, THE UNIT OF ANALYSIS IS “WOMEN-DAUGHTER PAIRS”. WE HAVE CORRECTED THIS AND ELSEWHERE IN THE MANUSCRIPT

2. Under line 167-168, the authors state “Data were weighted and statistical significance determined at 5%”. Can the authors indicate how data was weighted?

THE DHS DATA PROVIDED SAMPLING WEIGHTS FOR EACH PARTCIAPNT. WE APPLIED THE SAMPLING WEIGHTS IN THE ANALYSIS. THE PROCESS OF APPLYING THE WEIGHT IS OFTEN REFERRED TO AS WEIGHTING, SO THE DATA WERE WEIGHTED. WE HAVE CHANGED THE STATEMENT TO READ “WE APPLIED SAMPLING WEIGHTS TO THE DATA”

3. Can the authors indicate how they accounted for the complex sampling design of the surveys?

FIRSTLY WE USED INTRA-COUNTRY WEIGHTING WHICH HAD ALREADY ACCOUNTED FOR THE COMPLEXITY IN TERMES OF UNEQUAL POPULATION SIZES IN EACH CLUSTERS, REGIONS AND STATES. MORE IMPORTANTLY, WE USED CLUSTERED ANALYSIS THROUH MULTI-LEVEL MODELLING THAT ACCOUNTED FOR THE HIERACHICAL NATURE OF THE DATA

4. The authors pooled data from 12 African countries. Where is weighting? I do not mean the DHS weighting, but I mean your data weighting after pooling as you deal with multiple countries with wide variation in their population. You can refer to the reference below to understand what I mean.

Marriott, B. M., Campbell, L., Hirsch, E., & Wilson, D. (2007). Preliminary data from demographic and health surveys on infant feeding in 20 developing countries. The Journal of nutrition, 137(2), 518S-523S.

WE APPRECIATE THIS IMPORTANT ISSUE RAISED BY OUR REVIEWER. WE DID NOT ADJUST FOR THE DIFFERENCES IN SAMPLE SIZES OF EACH COUNTRY FOR THE FOLLOWING REASONS

a. MOST STATISTICAL SOFTWARE DO NOT MAKE PROVISION FOR INTER-COUNTRY WEIGHTING, RATHER WEIGHTING IS AVAILABLE FOR INTRA-COUNTRY WEIGHTING

b. WEIGHTING, EITHER INTER OR INTRA-COUNTRY, DOES NOT AFFECT MULTIVARIATE ANALYSIS EXCEPT FREQUENCIES

c. THE GOAL OF THE STUDY WAS NOT TO ESTIMATE AN AFRICAN-WIDE FIGURES AS 12 COUNTRIES CANNOT POSSIBLY REPRESENT OVER 50 COUNTRIES, SO WE WERE MORE CONCERNED ABOUT COUNTRY-SPECIFIC ANALYSIS FOR ALL THE UNIVARIATE AND BIVARIATE ANALYSIS WE PRESENTED. 

d. MARRIOTT ET AL ADJUSTED FOR VARIABILITY IN THE NUMBER OF INDIVIDUALS SAMPLED IN EACH COUNTRY AND NOT FOR POPULATION SIZES OF EACH COUNTRY.

e. MOST MULTI-COUNTRY STUDY DO NOT 

5. Under line 170-172, the authors state “Respondents with no female child and respondents with no information on whether the daughter was circumcised or not were excluded from further analysis.” This is an indication that there was missing data in each country’s dataset, can the authors provide the following information for each country

A. total women interviewed

B. Sample size by design

C. Selected women sample (this is already in Table 1).

D. Percentage with complete information on the inclusion criteria

E. Percentage of missing data.

THANK YOU. THIS STATEMENT DID NOT CONSTITUTE MISSING DATA. IT WILL BE MISLEADING TO CLASSIFY ALL SAMPLED WOMEN WITH NO RESPONSE TO DAUGHTERS CIRCUMCISION AS MISSING. FIRSTLY THERE ARE NO WAYS WOMEN WITH NO DAUGHTER CAN PROVIDE INFORMATION ON DAUGHTERS CIRCUMCISION OR OTHERWISE, SECONDLY, GOING BY THE SAMPLING DESIGN, WOMEN WITH NO LIVING DAUGHTER WERE AUTOMATICALLY EXCLUDED. THIRDLY, WOMEN FROM HOUSEHOLDS THAT WERE SELECTED FOR “MAN INTERVIEW” WERE EXCLUDED FROM “FEMALE GENITAL MUTILATION” MODULE. WE HAD REFERED HAVE TO THE MOTHERS IN THE SECOND AND THIRD CATEGORIES AS THOSE “RESPONDENTS WITH NO INFORMATION ON WHETHER THE DAUGHTER WAS CIRCUMCISED OR NOT”. 

THE SENTENCE HAVE BEEN REFRAMED TO READ “RESPONDENTS WHO DID NOT GIVE BIRTH TO ANY FEMALE CHILD, OR WHO HAS NO LIVING DAUGHTER WERE WAS DEAD OR WHO WERE SELECTED FOR MALE QUESTIONNAIRE ACCORDING TO THE SURVEY PROTOCOL WERE NOT ASKED QUESTIONS ON DAUGHTER’S CIRCUMCISION. OF THE 81,921 ASKED QUESTIONS ON DAUGHTER’S CIRCUMCISION, 80,403(98.1%) PROVIDED VALID RESPONSES (SUPPLEMENTARY TABLE B). THE REMAINING 1.9% WERE EXCLUDED FROM FURTHER ANALYSIS.”

DETAILS OF THE METHODOLOGY IS AVAILABLE AT https://dhsprogram.com/what-we-do/survey/survey-display-528.cfm

6. Finally, can the authors explain how they treated missing data?

ACCORDING TO DHS, THE SAMPLING WEIGHTS ARE COMPUTED AFTER DATA COLLECTION, SO THE SAMPLING WEIGHTS HAVE ACCOUNTED FOR NON RESPONSES AND UNEQUAL POPULATION SIZES IN EACH GEOGRAPHICAL DOMAINS. (https://dhsprogram.com/data/Guide-to-DHS-Statistics/index.htm#t=Analyzing_DHS_Data.htm). THE 1.9% INVALID RESPONSES WERE TREATED AS MISSING

 6. PLOS authors have the option to publish the peer review history of their article (what does this mean?). If published, this will include your full peer review and any attached files.

Do you want your identity to be public for this peer review? For information about this choice, including consent withdrawal, please see our Privacy Policy.

 Reviewer #1: Yes: Patrick O. Okoth

Reviewer #2: Yes: Bright Opoku Ahinkorah

 WE HAVE VERIFIED THE FIGURES AND UPLOADED ACCORDINGLY

Yours truly

ADENIYI FAGBAMIGBE

---

## [Decision Letter · Decision Letter 1]

20 Nov 2020

PONE-D-20-19500R1

Hierarchical modelling of factors associated with the practice and perpetuation of female genital mutilation in the next generation of women in Africa

PLOS ONE

Dear Dr. Fagbamigbe,

Thank you for submitting your manuscript to PLOS ONE. After careful consideration, we feel that it has merit but does not fully meet PLOS ONE’s publication criteria as it currently stands. Therefore, we invite you to submit a revised version of the manuscript that addresses the points raised during the review process.

We look forward to receiving your revised manuscript.

Kind regards,

Joshua Amo-Adjei, Ph.D

Academic Editor

PLOS ONE

Reviewers' comments:

Reviewer's Responses to Questions

**Comments to the Author**

1. If the authors have adequately addressed your comments raised in a previous round of review and you feel that this manuscript is now acceptable for publication, you may indicate that here to bypass the “Comments to the Author” section, enter your conflict of interest statement in the “Confidential to Editor” section, and submit your "Accept" recommendation.

Reviewer #2: All comments have been addressed

2. Is the manuscript technically sound, and do the data support the conclusions?

Reviewer #2: Yes

3. Has the statistical analysis been performed appropriately and rigorously? 

Reviewer #2: Yes

4. Have the authors made all data underlying the findings in their manuscript fully available?

Reviewer #2: Yes

5. Is the manuscript presented in an intelligible fashion and written in standard English?

Reviewer #2: Yes

6. Review Comments to the Author

Reviewer #2: I appreciate the efforts the authors have made in revising their manuscript per the comments of the editor and reviewers. I think all the issues I raised have been addressed. Although I consider the reasons the authors gave for not applying a weighting factor in the pooled data as not convincing, I accept their reasons in line with scholarly agreement.

Going through the revised paper, I have identified a statement which the authors made and which is not correct and needs to be addressed:

In line 98-99, the authors claimed “We identified only 12 countries with data set on FGM for both the respondents and their daughters.” This statement is not correct. This is because Sierra Leone is also a country in Africa with data set on FGM for both mother and daughter. In terms of prevalence of FGM in Africa, it is one of the countries with the highest prevalence of FGM (88%) (https://www.who.int/reproductivehealth/topics/fgm/prevalence/en/). I suggest the inclusion of Sierra Leone in this study as there seems to be no reason for leaving the country out.

7. PLOS authors have the option to publish the peer review history of their article (what does this mean?). If published, this will include your full peer review and any attached files.

Reviewer #2: **Yes: **Bright Opoku Ahinkorah

---

## [Author Response · Author response to Decision Letter 1]

24 Nov 2020

November 23rd 2020

Dear Editor

PLOS ONE 

Re: PONE-D-20-19500R1

Perpetration of female genital mutilation in the next generation of women from 12 African countries: Evidence from hierarchical modelling of their attributes

We the authors of above mentioned paper appreciate the efforts and comments of the editor and the eminent reviewers. We have addressed all these comments. A point-by-point response to the issues in our revised manuscript is listed below.

Please note that we used the file with the tracked changes to describe where the changes were made. 

PONE-D-20-19500R1

Hierarchical modelling of factors associated with the practice and perpetuation of female genital mutilation in the next generation of women in Africa

PLOS ONE

Dear Dr. Fagbamigbe,

Thank you for submitting your manuscript to PLOS ONE. After careful consideration, we feel that it has merit but does not fully meet PLOS ONE’s publication criteria as it currently stands. Therefore, we invite you to submit a revised version of the manuscript that addresses the points raised during the review process.

1. If the authors have adequately addressed your comments raised in a previous round of review and you feel that this manuscript is now acceptable for publication, you may indicate that here to bypass the “Comments to the Author” section, enter your conflict of interest statement in the “Confidential to Editor” section, and submit your "Accept" recommendation.

Reviewer #2: All comments have been addressed

THANK YOU

2. Is the manuscript technically sound, and do the data support the conclusions?

Reviewer #2: Yes

THANK YOU

3. Has the statistical analysis been performed appropriately and rigorously?

Reviewer #2: Yes

THANK YOU

4. Have the authors made all data underlying the findings in their manuscript fully available?

Reviewer #2: Yes

THANK YOU

5. Is the manuscript presented in an intelligible fashion and written in standard English?

Reviewer #2: Yes

THANK YOU

6. Review Comments to the Author

Reviewer #2: I appreciate the efforts the authors have made in revising their manuscript per the comments of the editor and reviewers. I think all the issues I raised have been addressed. Although I consider the reasons the authors gave for not applying a weighting factor in the pooled data as not convincing, I accept their reasons in line with scholarly agreement.

Going through the revised paper, I have identified a statement which the authors made and which is not correct and needs to be addressed:

In line 98-99, the authors claimed “We identified only 12 countries with data set on FGM for both the respondents and their daughters.” This statement is not correct. This is because Sierra Leone is also a country in Africa with data set on FGM for both mother and daughter. In terms of prevalence of FGM in Africa, it is one of the countries with the highest prevalence of FGM (88%) (https://www.who.int/reproductivehealth/topics/fgm/prevalence/en/). I suggest the inclusion of Sierra Leone in this study as there seems to be no reason for leaving the country out.

THANK YOU. WE TOTALLY AGREE WITH YOUR COMMENTS. WE HAD EXCLUDED SIERRA LEONE AND NIGER BECAUSE IT THE DATASET FOR THESE TWO COUNTRIES DID NOT CAPTURE SOME RELEVANT INFORMATION. 

WE HAVE NOW INCLUDED THE TWO COUNTRIES, REANALYSED THE DATA ENTIRELY AND UPDATED ALL RELEVANT SECTIONS OF THE PAPER. THE INCLUSION OF THESE COUNTRIES WILL DEFINITELY STRENGTHEN OUR FINDINGS.

7. PLOS authors have the option to publish the peer review history of their article (what does this mean?). If published, this will include your full peer review and any attached files.

Do you want your identity to be public for this peer review? For information about this choice, including consent withdrawal, please see our Privacy Policy.

Reviewer #2: Yes: Bright Opoku Ahinkorah

THANK YOU

---

## [Decision Letter · Decision Letter 2]

18 Mar 2021

PONE-D-20-19500R2

Hierarchical modelling of factors associated with the practice and perpetuation of female genital mutilation in the next generation of women in Africa

PLOS ONE

Dear Dr. Fagbamigbe,

Thank you for submitting your manuscript to PLOS ONE. After careful consideration, we feel that it has merit but does not fully meet PLOS ONE’s publication criteria as it currently stands. Therefore, we invite you to submit a revised version of the manuscript that addresses the points raised during the review process.

We look forward to receiving your revised manuscript.

Kind regards,

Susan A. Bartels, MD, MPH, FRCPC

Academic Editor

PLOS ONE

Journal Requirements:

Reviewers' comments:

Reviewer's Responses to Questions

**Comments to the Author**

1. If the authors have adequately addressed your comments raised in a previous round of review and you feel that this manuscript is now acceptable for publication, you may indicate that here to bypass the “Comments to the Author” section, enter your conflict of interest statement in the “Confidential to Editor” section, and submit your "Accept" recommendation.

Reviewer #2: All comments have been addressed

Reviewer #3: (No Response)

2. Is the manuscript technically sound, and do the data support the conclusions?

Reviewer #2: Yes

Reviewer #3: Yes

3. Has the statistical analysis been performed appropriately and rigorously? 

Reviewer #2: Yes

Reviewer #3: Yes

4. Have the authors made all data underlying the findings in their manuscript fully available?

Reviewer #2: Yes

Reviewer #3: Yes

5. Is the manuscript presented in an intelligible fashion and written in standard English?

Reviewer #2: Yes

Reviewer #3: Yes

6. Review Comments to the Author

Reviewer #2: Thank you for spending time to incorporate the suggestions. All comments have been addressed. Well done.

Reviewer #3: A research study was conducted to access the current practice of mother and daughter FGM (female genital mutilation). A hierarchical multivariable logistic regression model was fitted. The results indicate that the risk of mothers having FGM for their daughters was significantly associated with the following factors: maternal age, educational status, religion, household wealth quintiles, place of residence, community unemployment and community poverty.

Minor revisions:

1- Lines 135 and 192: Improve the clarity of the statement by revising these sentences.

2- Line 270: Revise “mean mothers’ age” to “mothers’ mean age.”

3- P-values never equal zero; express small p-values as p < 0.001.

4- Table 4: to conform to standard practice, provide only the overall p-value for each factor rather than p-values for each level of each factor.

7. PLOS authors have the option to publish the peer review history of their article (what does this mean?). If published, this will include your full peer review and any attached files.

Reviewer #2: **Yes: **Bright Opoku Ahinkorah

Reviewer #3: No

---

## [Author Response · Author response to Decision Letter 2]

18 Mar 2021

6. Review Comments to the Author

Reviewer #2: Thank you for spending time to incorporate the suggestions. All comments have been addressed. Well done.

Thank you

Reviewer #3: A research study was conducted to access the current practice of mother and daughter FGM (female genital mutilation). A hierarchical multivariable logistic regression model was fitted. The results indicate that the risk of mothers having FGM for their daughters was significantly associated with the following factors: maternal age, educational status, religion, household wealth quintiles, place of residence, community unemployment and community poverty.

Minor revisions:

1- Lines 135 and 192: Improve the clarity of the statement by revising these sentences.

Thank you. We have revised the sentences to read “We included only respondents (women) with response on at least one daughter’s FGM status in this study. This was to allow mother-daughter analysis of FGM practice. Our analysis may, therefore, differ slightly from the published estimates on level of FGM among women and children by the DHS for each of the countries.” 

and 

“The remaining 2.2% with invalid responses on FGM among daughters were excluded from further analysis.” respectively

2- Line 270: Revise “mean mothers’ age” to “mothers’ mean age.”

Thank you

3- P-values never equal zero; express small p-values as p < 0.001.

Thank you. We have changed all the appearances

4- Table 4: to conform to standard practice, provide only the overall p-value for each factor rather than p-values for each level of each factor.

Thank you. We appreciate your comment and would have been idea if the p-value is from a chi-square test. But this is not the case here, we carried out a logistic regression, where all estimate for each levels were compared with the reference level for each factor. It is possible that one level is significant while the other is not significant. For example, marital status in Fig 4: never married people had significant lower odds than the currently married people, but the formerly married people had insignificant odds compared with the currently married people. It will therefore be wrong to use a single p-value for the two. Therefore, the current presentation of p-values are correct.

---

## [Editor Report · Decision Letter 3]

7 Apr 2021

Hierarchical modelling of factors associated with the practice and perpetuation of female genital mutilation in the next generation of women in Africa

PONE-D-20-19500R3

Dear Dr. Fagbamigbe,

We’re pleased to inform you that your manuscript has been judged scientifically suitable for publication and will be formally accepted for publication once it meets all outstanding technical requirements.

Kind regards,

Susan A. Bartels, MD, MPH, FRCPC

Academic Editor

PLOS ONE
---

## [Editor Report · Acceptance letter]

12 Apr 2021

PONE-D-20-19500R3 

Hierarchical modelling of factors associated with the practice and perpetuation of female genital mutilation in the next generation of women in Africa 

Dear Dr. Fagbamigbe:

I'm pleased to inform you that your manuscript has been deemed suitable for publication in PLOS ONE. Congratulations! Your manuscript is now with our production department. 

Kind regards, 

on behalf of

Dr. Susan A. Bartels 

Academic Editor

PLOS ONE